# Who Will Be More Egocentric? Age Differences in the Impact of Retrospective Self-Experience on Interpersonal Emotion Intensity Judgment

**DOI:** 10.3390/bs14040299

**Published:** 2024-04-04

**Authors:** Menghan Jin, Huamao Peng

**Affiliations:** 1Institute of Developmental Psychology, Beijing Normal University, Beijing 100875, China; 202031061023@mail.bnu.edu.cn; 2Beijing Key Laboratory of Applied Experimental Psychology, Beijing Normal University, Beijing 100875, China

**Keywords:** egocentric, older adults, emotion judgment, anchoring effect

## Abstract

This study investigates whether the retrospective self-experience of older adults affects and biases interpersonal emotion judgment more than that of younger adults by adopting the paradigm of the self-generated anchoring effect. Participants (older adults: *n* = 63; younger adults: *n* = 65) were required to retrospectively consider their self-experiences and judge their possible emotion intensity in anchor-generating scenarios (high- or low-anchor scenarios). Subsequently, participants estimated the protagonist’s emotion intensity in target scenarios. The age-related interaction effect showed that older adults exhibited a significant self-generated anchoring effect in more emotion categories (four emotions) compared with younger adults (two emotions). After controlling for inhibition or working memory as a covariant, this interaction effect was no longer significant. The results from multilevel regression analysis also indicated the significant effect of self-emotion across all models on participants’ judgment of others’ emotions. The results indicated that older adults were more affected by retrospective self-experiences, leading to more egocentric judgment, than younger adults. This different influence from the retrospective self-experiences might partially have been caused by the age-related difference in cognitive abilities.

## 1. Introduction

Judging the emotional state of others involves a partially self-anchored or self-centered process. Individuals must first estimate their own emotional state under specific situations, then compare similarities and differences between themselves and others, and finally make judgments about others’ emotions based on their own possible emotional states [1]. In addition to exhibiting more egocentric thinking biases compared with younger adults [2], older adults tend to make more mistakes than their younger counterparts in mental state inference tasks and emotion judgment tasks [3]. This trend is likely influenced by the decrease in perspective-taking ability, including emotional perspective-taking, and in theory of mind skills as individuals age [3,4,5]. The anchoring effect was one of the far-reaching representations of judgment bias [6,7]. The current study sought to explore whether older adults perform more egocentric thinking bias in interpersonal emotion intensity judgment and explore the possible cognitive mechanism by using verbal scenarios as emotion judgment contexts connected with the traditional anchoring effect paradigm. 

### 1.1. Age Difference in the Impact of Self-Experience on Emotional Judgment

In everyday life, the ability to infer others’ emotions, also known as emotional perspective-taking, requires individuals to review their perspectives and emotional states in specific situations [1]. Evidence from an attitude-inferring study indicates that individuals anchor and adjust their judgments based on their perspective when inferring the opinions and attitudes of similar others [8]. Therefore, older adults’ inadequate performance in interpersonal emotion judgment might be related to their excessive reliance on self-perspective. 

#### 1.1.1. The Reasons of Age Difference in The Impact of Self-Experience

Firstly, older adults may encounter more difficulties in interpreting and predicting the beliefs, intentions, and emotions of others, leading to increased engagement in egocentric thinking. For example, they were found to engage in more egocentric thinking compared with younger adults, not only in egocentrism-related questionnaires but also in some false belief tasks [2], due to their decreased ability in perspective-taking, including emotional perspective-taking [4]. Thus, it would be more effective and convenient for older adults to rely more on their perspectives because past experiences could help to infer others’ possible emotional reactions to similar events. 

Secondly, older adults may exhibit a heightened focus on their own internal states or perspectives. This tendency is supported by research on age differences in off-topic verbosity, which revealed that older adults encounter greater challenges in restraining the interference of autobiographical information [9,10]. This suggests that older adults may be more susceptible to the influence of inner irrelevant information, potentially resulting in heightened tendencies toward egocentric judgment. 

Lastly, when we interpret these age differences in depth, inferring others’ thoughts and emotions is related to one’s declined cognitive abilities. Processing speed, working memory, and inhibition are widely recognized as important variables of cognitive aging [11]. The lack of cognitive resources may affect how older adults synthesize and analyze clues from contexts to effectively understand others [12]. Insufficient information processing may encourage them to use more mental shortcuts (e.g., egocentrism; heuristics) [5]. Past research evidence also indicated that cognitive aging affects the ability to infer the perspectives of others’ thoughts, feelings, and emotions [3,5]. Specifically, processing speed and working memory are crucial for immediate information processing. Reduced inhibition control may make older adults more vulnerable to irrelevant information and inner prepotent responses [13], because executive functions contribute to perspective-taking [14]. In summation, it may be inferred that older adults’ increased reliance on self-experience may be explained by the age-related decline in cognitive abilities. 

#### 1.1.2. Age Difference in Judging Positive and Negative Emotions

Moreover, previous research has indicated that older adults display a preference for positive information rather than negative information in attention and memory [15,16]. Similar experience has been proven to be helpful in empathy and perspective-taking tasks [17]. Additionally, older adults are more susceptible to the interference of contextual emotional stimuli, making them more prone to underperforming than younger adults when judging negative emotions [18]. Therefore, based on these findings, the present study speculates that, compared with younger adults, older adults exhibit more egocentric tendencies in the judgment of negative emotions than positive emotions.

### 1.2. Self-Experience and the Self-Generated Anchoring Effect

Anchoring effects occur when judgments are biased toward a specific value presented prior to the judgment, leading estimates to assimilate to the anchor [6]. The self-generated anchoring effect refers to the judgment bias tendency of individuals generating a specific value based on their own experience and acquired clues, which then affects their judgment [7,19]. Typically, the classical comparison–judgment two-step paradigm of the anchoring effect involves high- and low-anchor conditions. The presence of an anchoring effect is established when a significant difference in estimates between these conditions emerges, indicating that the anchors impact judgment [6]. Previous research has shown that experimental methods using virtual textual scenario stories have revealed how participants’ judgments of others’ emotions (protagonists in the stories) are influenced by anchoring effects [20,21]. Specifically, research has found that presenting participants with judgment exemplars containing high or low anchor values influences the final emotional judgments for the protagonists [20]. Additionally, studies have shown that when participants are presented with textual scenarios, both older and younger adults compare the emotional intensity of the protagonists with the anchor value (high or low) before making judgments or estimates. Consequently, both older and younger adults are influenced by the anchor value, exhibiting significant anchoring effects [21].

The self-generated anchoring effect was used to explain the egocentric phenomenon of perspective-taking and social inferences [22]. In the study of Epley et al. [23], after being informed or not informed of the anchor background information, participants were asked to infer an individual’s understanding of an ambiguous phone recording, despite that individual not having the same background information that could bias their interpretation of the recording. The results revealed that the anchor background information influenced the participants’ judgment of the other’s opinions. They suggested that the egocentric phenomenon in perspective-taking could be explained by individuals’ insufficient adjustment from the anchored self-perspectives [19,23]. Thus, the self-generated anchoring effect paradigm is well suited to investigating the impact of retrospective self-experience on egocentric judgment.

### 1.3. Overview of the Present Study

#### 1.3.1. The Limitations of Past Research and the Implications of the Current Study

To date, the egocentric tendency of older adults in interpersonal judgment has been extensively discussed in the literature [2,24,25], and the compensating role of experience for older adults has been considered advantageous [26]. Despite its potentially extensive influence on their daily lives, the adverse effects of retrospective self-experience on older adults have been infrequently studied in the literature. Furthermore, the self-generated anchoring effect paradigm has seldom been used directly to explore the impact of retrospective self-experience. In the study by Epley et al. [23], the anchor information pertained to extra context cues related to the judgment target, rather than self-experience. Similarly, in research on the anchoring effect of emotion judgment [20], the anchoring information was provided by the experimenter and did not originate from the participants. Therefore, if the anchors are derived from participants’ own experiences, the role of retrospective self-experience in emotion judgment can be better represented.

This study investigates, for the first time, the potential for older adults’ retrospective self-experience to result in biased interpersonal judgment. Moreover, this study expands the development period to older adults for this important judgment bias. The study focuses on the specific role of self-experience in older adults and explores the cognitive aging mechanisms underlying the egocentric tendency of older adults. This could lead to a better understanding of the role of enriched experience in older adults’ judgment and decision-making, a double-edged sword of a role.

#### 1.3.2. The Logic of the Present Study and the Hypotheses

The current study utilized novel experimental designs, including life-like emotion scenarios, to better reflect emotion judgment in daily life. The participants were first presented with an anchor-generating scenario that described an emotional interpersonal event. They then retrospectively recalled their own related experiences according to the scenario and judged their possible emotion intensity in that scenario. This self-emotion intensity estimate represented their self-experience under that situation and provided the self-generated anchor for the subsequent judgment. After this anchor-generating phase, the participants estimated the emotion intensity of the target protagonist in a similar (event-type-like) scenario. Thus, the emotion judgment of others might be anchored on their retrospective experiences and corresponding intensity ratings. Moreover, cognitive abilities were measured to better understand the mechanism behind the impact of anchored self-experience on age differences in egocentric emotion judgments. 

In this study, the core focus is on the anchoring effect, which is represented by the significant distance in participants’ ratings of target emotions between high- and low-anchor conditions. Individual judgments of target emotions in the task are hypothesized to be influenced by participants’ ratings of their own emotions (self-emotion) under the anchoring conditions, as indicated by our assumptions.

As shown in Figure 1, ratings of self-emotion are directly influenced by individuals’ retrospective experiences. The adjustment from self-emotion ratings to ratings of the target person’s emotions involves cognitive processes, such as cognitive resources (processing speed and working memory), which may affect individuals’ cognitive effort and the choice of heuristic judgment strategies. Additionally, individuals need cognitive control to suppress the excessive influence of self-experiences, indicating a possible impact of inhibitory control abilities.

The hypotheses of the present study are illustrated below.

**Hypothesis** **1.**
*Retrospective self-experience (self-generated anchor) has a disproportionate impact on emotion judgment in older adults; older adults show more egocentric emotion judgment, i.e., are more affected by the anchor, than younger adults.*


**Hypothesis** **2.**
*The impact of retrospective self-experience on emotion judgment is stronger for negative emotion categories (anger, distress, and sadness) than for positive emotion categories (joy and pride) in older adults.*


**Hypothesis** **3.**
*Age differences in the impact of retrospective self-experience can be explained by differences in cognitive abilities. Specifically, after controlling for cognitive abilities, the age-related differences in the anchoring effect will be reduced or eliminated.*


## 2. Method

### 2.1. Participants

This study recruited 63 older adults (age: *M* = 64.03, *SD* = 3.66, range: 60–79; 30 males) and 65 younger adults (age: *M* = 23.20, *SD* = 3.28, range: 18–33; 28 males). The sample size was decided using G*power 3.1 [27]. The F-test for an ANOVA with repeated measures and between factors, had a typical medium size f of 0.25 (the referential effect size from Yit et al. was excessively large with an f above 0.57 [20], which resulted in an extremely low calculated sample size; thus, a medium effect size was chosen), alpha of 0.05, and power of 0.90. Furthermore, four groups and 10 measurements (see details in the study design) were selected. The calculated sample size indicated that 132 participants were required. Accordingly, approximately 30 participants were recruited for each condition. The calculated achieved power with a final sample size of 128 was 0.90. Older adults were recruited from communities in Beijing, and younger adults were mainly recruited from Beijing Normal University and other universities in Beijing.

### 2.2. Study Design

This study adopted a 2 (age group: old, young) × 2 (anchor: high, low) × 2 (emotional valence: positive, negative) mixed design, with age group and anchor as between-subject variables and emotional valence as a within-subject variable. 

In the experiment, all participants were required to read and imagine an emotion scenario about themselves (e.g., your child disappointed you because they don’t want to work). In the anchor-generating phase, they had to retrospectively recall a related self-experience and judge their possible emotion intensity on a 0 (not at all) to 100 (extremely) scale. Afterward, they had to read a similar scenario, the target scenario (e.g., Mr Wang’s kid disappointed him because they don’t want to get married), and judge the protagonist’s (e.g., Mr Wang’s) emotion intensity. The procedure of the experimental task is illustrated in Figure 2. 

The dependent variable index was the participants’ numerical estimate of the emotion intensity of the protagonist in the target scenario. Based on a previous study about the anchoring effect [6], the anchoring effect index was defined as the significant difference in the mean estimates for the target scenario between the high- and low-anchor conditions. A higher difference indicates a greater anchoring effect. The anchor-generating scenario was manipulated as a high- or low-anchor condition, as explained in the following part of the interpersonal emotion judgment task. Emotional valence represents the valence of the emotions felt by the protagonist in the target scenario, including positive emotions (joy and pride) and negative emotions (anger, distress, and sadness). 

### 2.3. Interpersonal Emotion Judgment Task

In formal experiments, ten anchor-generating scenarios were employed for both older and younger adults. Within these ten scenarios, each emotional category comprised two trials. The scenario materials used in the experiment were selected from a material pool created in the pilot study, which collected daily life emotional events from interviews with younger and older adults. Varying interpersonal scenarios for both older and younger adults were compiled due to their distinct life experiences. For example, scenarios such as “children returning home on vacation” and “obtaining a scholarship” corresponded to older and younger adults’ emotions of joy, respectively. This task bears resemblance to the task employed in our previous study [21], albeit with modifications tailored to address distinct research inquiries.

The selection of emotion categories was based on their frequency in interviews to ensure their universality and representativeness for both age groups. Joy and pride were chosen to represent positive emotions, while anger, distress, and sadness were selected to represent negative emotions. Once the emotion categories were determined, emotional scenario themes were identified based on their frequency in interviews. One or two scenario themes for each emotion category, which were mentioned frequently in interviews, were selected. For instance, for older adults, one of the selected joy-related scenario themes was “interpersonal interaction” (frequency ratio: 32.50%; e.g., visits from children, meetings with friends). Specific emotion scenarios for both younger and older adults were then compiled by researchers based on the selected scenario themes. These scenarios were screened based on ratings provided by a calibration group, which assessed their familiarity, importance, and emotional intensity. This process resulted in the generation of the scenario pool.

The anchor-generating scenarios and the target scenario materials were all chosen and edited from the scenario pool generated in the pilot study according to the emotion intensity rating results of each scenario from a calibration group (sixty-two older adults, age: *M* = 64.18; *SD* = 4.01; and fifty-five younger adults, age: *M* = 22.62; *SD* = 3.30). To ensure that the anchor-generating scenarios induced high and low emotional experiences in the high- and low-anchor conditions, the scenarios were selected from the calibration group ratings, with those higher than 75 chosen for the high-anchor condition and those lower than 65 for the low-anchor condition. Scenarios with ratings between 65 and 75 were selected for the target scenario. The calibration group also rated the familiarity and importance of the scenario (on a seven-point Likert scale from 1 = not familiar/important at all to 7 = extremely familiar/important) to ensure the consistency of experimental scenario materials for both age groups. The ratings of the scenario materials by the calibration group showed no significant age-related differences in familiarity and importance. Meanwhile, emotion categories were balanced in anchor-generating and target scenarios. The mean emotion intensity rating of target scenarios and the mean rating of the anchor-generating scenario for the high- and low-anchor conditions from the calibration group are presented in Appendix A. Furthermore, the detailed process of the pilot study is presented in Appendix A.

The final ten anchor-generating scenarios were selected and used in formal experiments for older and younger adults. Moreover, the calibration group’s mean ratings of the anchor-generating scenario for the high-anchor condition were significantly higher than that of the low-anchor condition, *t*(110) = 8.46, *p* = 0.000, which verified the validity of manipulation on the anchor. Similarly, ten target scenarios were selected for older and younger adults (no significant difference for age groups; *t*(122) = 0.16; *p* = 0.87). The two anchor conditions shared different anchor-generating scenarios but had the same target scenarios. The samples of task material in the high- and low-anchor conditions are shown in Table 1.

### 2.4. Other Measures

#### 2.4.1. Demographic Information

Demographic information, including age, gender, education, self-rated health condition, and family income (yuan/per month) was measured in the questionnaire.

#### 2.4.2. Cognitive Abilities

Processing speed was measured using the digit comparison task [28]. Participants judged whether or not a pair of digit strings were identical (e.g., 658331-656331) in the allotted time (90 s). The score was measured by the maximum number of items completed correctly. 

Working memory was measured using the backward digit span task from the Wechsler Adult Intelligence Scale (WAIS), 3rd edition [29]. Participants were asked to recite the digit strings, which they had just heard, backward. Working memory span was measured by the maximum length one could repeat correctly.

Inhibition ability was measured by the Stroop paradigm [30]. The E-prime program was used to perform the test, which required participants to identify the color of the displayed Chinese color characters. The difference in the average reaction time in the word color inconsistent condition minus that in the word color consistent condition was the index of inhibition ability. A high difference in reaction time meant low inhibition.

### 2.5. Procedure

As shown in Figure 2, participants provided their written informed consent before the experiment. They then completed a demographic information questionnaire and measures of cognitive abilities. Next, a baseline emotion intensity judgment task was completed by all participants to ensure that the judging tendency in the two groups (the high- and low-anchor group) had no systematic difference. The baseline task and the corresponding results are presented in Appendix A.

Lastly, older and younger participants were randomly assigned to the high- or low-anchor group. During the experiment, participants were required to complete one practice task and the formal experimental tasks. To balance the order effect, two sequences of tasks were produced. Half of the participants received materials in Sequence 1, which began with positive emotional trials, and the other half received materials in Sequence 2, which began with negative emotional trials. 

### 2.6. Analysis Plan

The data were initially analyzed via an analysis of variance (ANOVA) using SPSS.26 with repeated measures to detect main effects and interactions. Then, cognitive abilities were controlled as covariates in further analyses to test their role in explaining the age difference in the egocentric performance of emotion judgment, using the analysis of covariance (ANCOVA) in SPSS.26. 

Finally, in order to investigate the influence of self-emotion judgment on the perception of others’ emotions from a within-subject perspective, as well as its interaction with age group, multilevel modeling was developed utilizing the statistical software R. These models were designed to analyze the data collected from each participant, who engaged in five distinct categories of emotional judgment tasks.

The significance level was set at α = 0.05 for all statistical inferences.

## 3. Results

### 3.1. Basic Information of Participants

The descriptive statistics results of background variables are presented in Table 2. The results indicated that participants in the high- and low-anchor groups were generally similar in terms of these background variables (no significant differences), while older adults tend to have fewer years of education (*t*(121) = 11.61, *p* = 0.000), slower processing speed (*t*(126) = 17.78, *p* = 0.000), a smaller working memory span (*t*(125) = 10.08, *p* = 0.000), and lower inhibition ability (*t*(119) = 2.78, *p* = 0.006), compared with younger adults. The Pearson correlation tables depicting the relationships between measured variables for each age group in the present study are presented Supplement C.

### 3.2. Manipulation Check

The mean rating of participants’ self-emotion judgments for the anchor-generating scenario in the high- and low-anchor groups were as follows: *M*_high_ = 84.67 ± 9.72; *M*_low_ = 61.86 ± 15.04. A 2 (age: young, old) × 2 (anchor: high, low), ANOVA was conducted on the mean estimates of self-emotion judgments. The main effect of anchor was significant, with *F*(1, 124) =102.33, *p* = 0.000, and η_p_^2^ = 0.452, which verified the successful manipulation. No other significant main effect or interaction effect was found. Specifically, for the main effect of age, the analysis yielded a non-significant result: *F*(1, 124) = 0.01, *p* = 0.941, and η_p_^2^ = 0. Similarly, the interaction effect between age and anchor was not statistically significant: *F*(1, 124) = 0.33, *p* = 0.569, and η_p_^2^ = 0.003.

### 3.3. Anchoring Effect Analysis on Mean Estimates

A 2 (age: young, old) × 2 (anchor: high, low) × 2 (emotional valence: positive, negative) repeated measures ANOVA was conducted on the mean estimates of emotion intensity. The expected interaction between valence, age, and anchor was not found. However, the impact of the emotion category was shown as we replaced the emotional valence with the emotion category. To better explore the resulting pattern, the emotion category was adopted to replace emotional valence in subsequent analyses. Furthermore, years of education should be considered a covariate in the analysis because of the significant difference between the two age groups. The ANCOVA was conducted according to suggestions from Schneider et al. [31], and no significant effects of covariate were found; these are presented in Appendix A. Subsequently, A 2 (age: young, old) × 2 (anchor: high, low) × 5 (emotion category: joy, pride, anger, distress, sadness) repeated measures ANOVA was also conducted on the mean estimates to evaluate all other effects. The descriptive statistics are presented in Table 3. The results verified the significant anchoring effect (main effect of anchor): the mean estimates in the high-anchor condition were significantly higher than those in the low-anchor condition. *M*_high_ = 78.52 ± 10.45, *M*_low_ = 70.20 ±12.94, *F*(1, 119) = 16.04, *p* = 0.000, and η_p_^2^ = 0.119. The interaction effect between the emotion category, anchor, and age was significant, with *F*(4, 116) =2.57, *p* = 0.041, and η_p_^2^= 0.081. Results from simple effect analysis showed that age group differences in the anchoring effect perform differently according to different emotion categories. In general, older adults performed with a significant anchoring effect for four emotion categories (joy: *F*(1, 119) = 4.82, *p* = 0.030, *η_p_^2^
*= 0.039; pride: *F*(1, 119) = 10.57, *p* = 0.001, η_p_^2^ = 0.082; distress: *F*(1, 119) = 12.06, *p* = 0.001, η_p_^2^ = 0.092; sadness: *F*(1, 119) = 7.54, *p* = 0.007, η_p_^2^ = 0.060). However, younger adults only performed with a significant anchoring effect for two emotion categories (pride: *F*(1, 119) = 4.28, *p* = 0.041, η_p_^2^ = 0.035; anger: *F*(1, 119) = 4.79, *p* = 0.031, η_p_^2^ = 0.039). Both age groups performed with a significant anchoring effect on pride judgment, and older adults showed a larger size effect (older: η_p_^2^ = 0.082; younger: η_p_^2^ = 0.035). Besides these, the main effect of the emotion category was also significant, with *F*(4, 116) = 13.05, *p* = 0.000, and η_p_^2^ = 0.310. The mean difference in estimates between high- and low-anchor conditions and the corresponding significance are presented in Table 3. No other significant main effect or interaction effect was found. 

To further explain the interaction, three cognitive variables (being centered as recommended by Schneider et al. [31]) were controlled as a covariate in ANCOVAs. The results showed that the three-way interaction effect was no longer significant after inhibition was controlled as a covariate (*F*(4, 104) = 2.17, *p* = 0.078, and η_p_^2^ = 0.077), and was marginally significant after working memory was controlled (*F*(4, 114) = 2.46, *p* = 0.05, and η_p_^2^ = 0.079). However, it was still significant after processing speed was controlled (*F*(4, 115) = 2.79, *p* = 0.030, and η_p_^2^ = 0.088). Further simple effect analyses were conducted. The mean difference in estimates between high- and low-anchor conditions is presented in Table 3. As seen from these results, the mean difference and the corresponding effect size were decreased for older adults in three emotion categories (pride, anger, and distress), and increased for younger adults in three emotion categories (anger, distress, and sadness) after inhibition was controlled. Similarly, the mean difference and the corresponding effect size were decreased for older adults in all the emotion categories and increased for younger adults in four emotion categories (pride, anger, distress, and sadness) after working memory was controlled. These findings suggest that the anchoring effect in older adults may be more pronounced, partially due to differences in cognitive abilities, particularly inhibition and working memory. However, the significance of other main effects and interaction effects remained the same after inhibition or working memory was controlled.

Despite the confirmation of our hypothesis regarding the influence of age group through the repeated-measures ANOVA, the limited number of trials per emotion category (only two) raises concerns about the robustness of the results. Additionally, considering the within-subject nature of the emotion categories, a multilevel modeling approach would provide a better understanding of the impact of self-emotion on the perception of others’ emotions from an intraindividual perspective, as well as the interaction with the age group. To address these concerns, we employed multilevel regression analysis using the statistical software R (version 4.1.3). Specifically, we nested the self-emotion judgment data of the five-emotion category tasks for each participant (level 1: within-subject) within the individual-level data (level 2: between-subject) while incorporating the age group as a level 2 predictor. The analysis focused on the mean estimates of target emotion, with the mean estimates of self-emotion (level 1), age group (level 2), and the interaction between the self-emotion and age group (cross-level) as predictors. Three models were conducted, gradually introducing new predictors, and the results are presented in Table 4. The findings consistently demonstrated a significant effect of self-emotion across all models, suggesting that participants’ retrospective self-experiences could indeed bias their judgment of others’ emotions. Furthermore, the interaction between self-emotion and age group was also significant, indicating that the egocentric impact of retrospective self-experiences is more pronounced in older adults (*β*_old_ = 0.56, *p* < 0.001; *β*_young_ = 0.44, *p* < 0.001), further supporting our initial hypothesis. The cognitive abilities were excluded from this analysis due to the presence of multicollinearity between age group and self-emotion estimates, which would have arisen had the cognitive abilities been introduced into the model.

## 4. Discussion

This study adopted the interpersonal emotion intensity judgment task to explain the egocentric judgment bias of younger and older adults. The results confirmed that both age groups were affected by their own retrospective self-experiences when judging others’ emotions. The significant three-way interaction effect observed in the ANOVA analysis, and the significant two-way interaction effect found in the multilevel modeling, both suggest older adults exhibited a more significant anchoring effect than younger adults did. This finding suggested that older adults’ judgments of others’ emotions were more likely to be impacted by their retrospective self-experience, even when they were asked to provide objective estimates. This supported Hypothesis 1. These results were consistent with previous studies on perspective-taking [23] and emotion judgment [20]. Epley et al. demonstrated that individuals’ self-generated anchoring effect could bias interpersonal judgment, although their focus was on personality judgment rather than emotion [23]. Similarly, Yik et al. confirmed the existence and influence of anchoring effects in interpersonal emotion judgment, despite their exclusive focus on experimentally provided anchors [20]. However, unlike these two studies, the present study further revealed that retrospective self-experience affected participants’ judgments of others’ emotions, even when the anchor-generating scenario and the target scenario were distinct events. 

Hypothesis 2 regarding emotional valence was not supported in the present study. Nevertheless, the overall task exhibited an age-related tendency. Older adults demonstrated significant anchoring effects in four emotional task categories, while younger adults only showed significant anchoring effects in two categories. These findings may suggest that older adults were more consistently and significantly affected by their retrospective self-experiences when assessing the emotions of others. Additionally, the age difference observed in this study contributes to the existing conclusions about older adults’ limited emotional perception and judgment accuracy in past research [4,18]. However, this age difference might also be affected by the emotion category. Specifically, the results for anger exhibited the most unique age difference, with older adults demonstrating better emotional judgment performance: the egocentric anchoring effect was not significant among older adults, whereas it was significant among younger adults. Previous research has suggested that older adults often struggle with recognizing negative emotions such as anger or sadness [32]. This finding suggests that the extent of egocentric anchoring effects in judging others’ emotions may not solely depend on the ability to perceive emotions. Instead, it may also be influenced by one’s own emotional experiences and sensitivity to emotional cues. Older adults were found to typically experience less anger daily than younger adults [33]. Accordingly, their sensitivity to anger diminishes with age during late adulthood, whereas sensitivity experiences the most pronounced increase with age among adolescents and younger adults [34]. Consequently, older adults’ anger emotions may be more stable due to reduced sensitivity to situations that could evoke anger emotions. Thus, their self-rated anger emotions under both high- and low-anchoring conditions may be closer than those of younger adults (self-emotion rating: M_high anchor_ = 73.67 ± 23.24; M_low anchor_ = 62.34 ± 22.72), leading to closer judgments of the target person’s anger emotions in two anchor conditions (target emotion rating: M_high anchor_ = 78.00 ± 12.52; M_low anchor_ = 72.37 ± 16.74). Conversely, younger adults’ sensitivity to anger emotions peaks, and their anger emotions may fluctuate more, making them more attuned to anger-related cues. Consequently, their self-anger emotions under both high- and low-anchoring conditions may vary more (self-emotion rating: M_high anchor_ = 82.16 ± 14.70; M_low anchor_ = 55.44 ± 18.85), leading to greater differences in judgments of the target person’s anger emotions in two anchor conditions (target emotion rating: M_high anchor_ = 78.31 ± 13.62; M_low anchor_ = 70.00 ± 17.46). Importantly, the inconsistent emotion category results regarding the age difference might also be partially due to our limited number of trials for each emotion category. Thus, the impact of emotional valence and category should not be drawn assertively based solely on the present results. 

Hypothesis 3 was supported in the present study, revealing that working memory and inhibition play a significant role in the three-way interaction effect (anchor condition, age group, and emotion category) on mean estimates. Specifically, after controlling for inhibition or working memory, the mean estimate difference—which represents the degree of the anchoring effect—and the corresponding effect size were reduced for older adults. In contrast, for younger adults, the mean estimate difference and the corresponding effect size increased in many emotion categories. As a result, the final three-way interactions were not significant, indicating that the age-related difference in the anchoring effect for these emotions was partially reduced. The impact of retrospective self-experience on egocentric emotion judgment in older adults may, therefore, partially result from the age difference in inhibition and working memory. The default mode of self-perspective is driven by the automatic link between perception and action [35], which is reflected in the anchoring effect as a kind of heuristic processing [7], where mental shortcuts are used instead of the effortful deliberation of facts [7,36]. Older adults may rely more on this heuristic processing because of their increased experience and declined cognitive abilities [37]. Specifically, inferring others’ perspectives or emotions requires an inhibition of self-perspective. Many studies indicated that older adults’ declined theory of mind might result from their limited executive inhibition [3,38]. Further, a decreased working memory span might limit older adults’ ability to fully maintain and utilize the information required to generate an appropriate judgment. Age-related impairment in inferring others’ thoughts and feelings might also result from older adults’ difficulties in inhibiting irrelevant information, and in information maintenance and extraction [39,40]. Processing speed was not a factor that explained the age-related difference in the present study, likely because the task was not time-limited, and, therefore, relied relatively less on processing speed. In summary, limited working memory and inhibition might lead older adults to process information ineffectively. Consequently, this may inhibit their feelings and emotions generated from the retrospective self-experience, leading to an egocentric judgment of others’ emotions. 

### 4.1. Strengths, Limitations, and Future Research

This study found that older adults are more susceptible to the anchoring effect in interpersonal emotion intensity judgment when influenced by retrospective self-experience. Additionally, working memory and inhibition play vital roles in this process. This study not only delved deeper into the biased and egocentric phenomenon in interpersonal emotion judgment but also revealed age-related characteristics with a unique manipulation of retrospective self-experience. Additionally, the results could have implications for daily interpersonal perception, judgment, and decision-making. Firstly, by recognizing the egocentric emotional perspective-taking tendency among older adults, we can promote better communication strategies, thereby enhancing interpersonal understanding and support. Secondly, understanding how older adults’ emotional judgments are influenced by their own emotional experiences can lead to more personalized and effective mental health interventions, improving the quality of mental health services for this demographic.

However, this study has several limitations that future research should consider. Firstly, the emotion categories were based on interviews in the pilot study, which aimed to represent daily emotions. Nevertheless, both primary emotions (joy, anger, and sadness) and secondary emotions (pride and distress) were included, and the latter might have different perception characteristics across cultures. Thus, caution should be taken regarding the generalization of the results. 

Secondly, this study had limited emotion categories for two emotional valences, which could not capture the characteristics of the valence. Future research could focus on specific emotion categories using a more elaborate design and could explore the effects of emotional valence in more depth. Furthermore, research has offered neurobiological evidence suggesting that healthy older adults effectively manage regret, displaying diminished responsiveness to “missed opportunities” due to their inclination to prioritize positive experiences and disengage from negative ones [41]. Similar mechanisms may be at play in older adults’ processing of other negative emotions. However, our study has not yet produced consistent and reliable findings concerning emotional valence and categories, underscoring the need for further investigation in future studies.

Finally, the examination of the role of cognitive abilities in this study was exploratory and conducted using a relatively rough analytic approach. Therefore, caution is warranted when interpreting the corresponding conclusions. Future research should employ more sophisticated analytic approaches, which will contribute to a more comprehensive understanding of age differences.

### 4.2. Conclusions

The current study demonstrated that older adults were more prone to egocentric emotional perspective-taking, as evidenced by the greater influence of their own emotional experience on judgments of others’ emotions. Differences in inhibition and working memory between the age groups partly explained the age-related differences in egocentric emotion judgment. In everyday life, to counteract this egocentric tendency, we should exercise caution when perceiving others’ emotions and consider differences between ourselves and others. Furthermore, older adults may benefit from taking more time to consider the situation and others’ perspectives before making judgments about their emotions.

## Figures and Tables

**Figure 1 behavsci-14-00299-f001:**
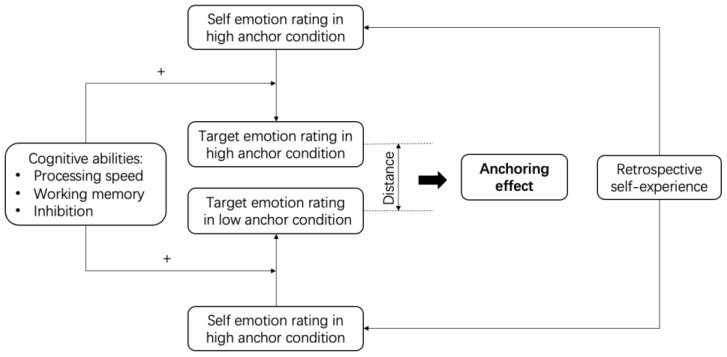
The logic of the present study. Note: The plus sign (+) between cognitive ability and self-emotion rating and target emotion rating in the figure indicates that the degree of adjustment from self-emotion rating to target emotion rating may be positively influenced by cognitive ability. The higher the cognitive ability, the greater the potential for adjustment, thus potentially weakening the anchoring effect demonstrated.

**Figure 2 behavsci-14-00299-f002:**
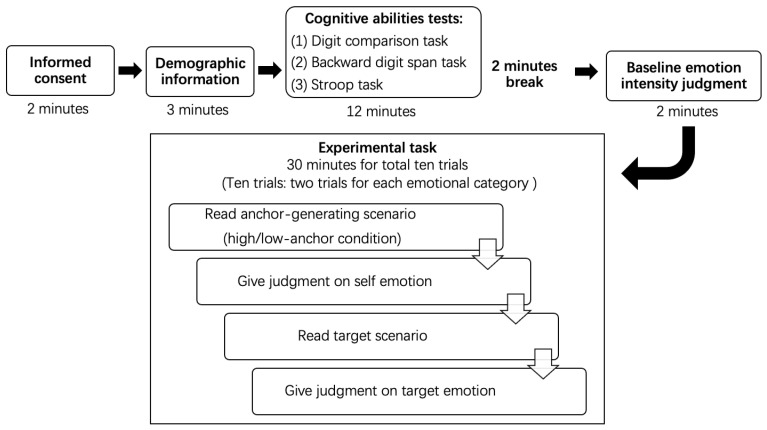
The experimental task procedure.

**Table 1 behavsci-14-00299-t001:** Sample of the experimental task for the high anchor and low anchor conditions (older adults).

	High Anchor Material Sample	Low Anchor Material Sample
Anchor-generating scenario	Your 30 years old son has been living in your house for years without looking for a job, and yesterday he complained that he did not want to go to work.	You don’t like your wife’s nagging. However, today you accidentally bought something wrong and your wife nagged you again.
Self-emotion judgment	Please retrospect your own related experience and what is the distress intensity you might feel in the above situation.	Please retrospect your own related experience and what is the distress intensity you might feel in the above situation.
Target scenario	Mr. Wang is over 60. He particularly hopes that his only daughter could get married as soon as possible. Yesterday, his daughter complained to him that she insisted on not going to the blind date.	Mr. Wang is over 60. He particularly hopes that his only daughter could get married as soon as possible. Yesterday, his daughter complained to him that she insisted on not going on the blind date.
Target emotion judgment	In your opinion, what is the distress intensity felt by the protagonist in the scenario?	In your opinion, what is the distress intensity felt by the protagonist in the scenario?

**Table 2 behavsci-14-00299-t002:** Description of the demographic variables of younger and older adults.

	High Anchor Group (N = 64)				Low Anchor Group (N = 64)			
	Old (N = 32)		Young (N = 32)		Old (N = 31)		Young (N = 33)	
	*M*	*SD*	*M*	*SD*	*M*	*SD*	*M*	*SD*
Age	64.00	4.03	23.41	3.27	64.06	3.30	23.00	3.33
Income (yuan/month)	8053.13	3066.89	11,366.67	7107.47	12,040.00	8821.51	17,419.35	26,338.53
Education (year)	11.37	2.76	16.76	2.44	11.00	2.33	15.72	2.09
Health	3.67	0.61	4.00	0.71	3.97	0.50	3.80	0.71
PS	20.68	6.81	42.16	6.10	21.90	7.03	42.45	7.03
WM	4.77	1.38	7.41	1.58	4.84	1.53	7.76	1.71
Stroop	190.41	185.59	95.84	75.34	121.86	160.00	81.07	68.82

Note: PS refers to processing speed. WM refers to working memory. Health refers to a self-rated health condition and is rated on a 5-point scale (1 = very poor; 2 = poor; 3 = average; 4 = good; and 5 = very good).

**Table 3 behavsci-14-00299-t003:** The mean emotion estimates of each emotion and the corresponding anchoring effect.

Emotion Category	Age Group	Mean Estimate of Emotion Intensity		Anchoring Effect: Mean Estimate Difference D(η_p_^2^)		
		High-Anchor *M* (*SD*)	Low-Anchor *M* (*SD*)	Original	Inhibition-Controlled	WM-Controlled
Joy	Old	79.59 (12.29)	69.83 (17.92)	9.76 * (0.039)	10.27 (0.053)	**9.74 (0.038)**
	Young	73.25 (15.68)	66.39 (20.17)	6.86 (0.022)	4.24 (0.010)	6.73 (0.021)
Pride	Old	87.21(10.26)	74.92(18.06)	12.30 ** (0.082)	**11.25 (0.073)**	**11.83 (0.075)**
	Young	82.42 (14.38)	75.03 (13.63)	7.39 * (0.035)	6.53 (0.027)	**7.44 (0.035)**
Anger	Old	78.00 (12.52)	72.37 (16.74)	5.63 (0.016)	**5.38 (0.015)**	**5.01 (0.013)**
	Young	78.31 (13.62)	70.00 (17.46)	8.31 * (0.039)	**9.50 (0.046)**	**8.49 (0.041)**
Distress	Old	78.75 (13.11)	63.48 (21.87)	15.27 ** (0.092)	**14.34 (0.080)**	**14.68 (0.086)**
	Young	73.25 (16.10)	71.64 (14.54)	1.61 (0.001)	**2.83 (0.004)**	**2.02 (0.002)**
Sadness	Old	77.23 (12.37)	66.22 (18.59)	11.02 ** (0.060)	11.46 (0.068)	**10.36 (0.053)**
	Young	79.39 (13.95)	72.55 (15.35)	6.85 (0.027)	**7.37 (0.030)**	**6.96 (0.028)**

Note: The difference in estimates (D) is shown in the column “Anchoring effect”, with η_p_^2^ in brackets. The column “original” presents the difference in estimates without cognitive ability being controlled. The significant anchoring effect is marked, in which “*” means *p* < 0.05; “**” means *p* < 0.01. The column “Inhibition controlled” and “WM controlled” refers to the difference in estimates after inhibition or working memory was controlled. The difference highlighted in bold means it changed with the age-related decrease direction after cognitive ability was controlled (decreased for older adults or increased for younger adults).

**Table 4 behavsci-14-00299-t004:** The multilevel regression models result in predicting the mean estimates of other’s emotion intensity.

Observations: 637	Model 1			Model 2			Model 3		
Predictor	*β*	*t*	*p*	*β*	*t*	*p*	*β*	*t*	*p*
(intercept)	73.77	106.87	<0.01	74.32	76.65	<0.01	74.32	76.91	<0.01
Self-emotion	**0.51**	**19.54**	**<0.01**	**0.51**	**19.53**	**<0.01**	**0.45**	**11.30**	**<0.01**
Age group				−1.13	−0.82	0.41	−1.13	−0.82	0.41
Self-emotion × Age group							**0.11**	**2.02**	**0.04**
σ^2^	123.11			123.10			122.57		
τ_00_	36.24			36.40			36.10		
ICC	0.23			0.23			0.23		
Marginal R^2^/Conditional R^2^	0.425/0.556			0.425/0.557			0.428/0.558		

Note: The results of significant predictors are highlighted in bold.

## Data Availability

The data presented in this study are available on request from the corresponding author.

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
