# Peer review of "Who Will Be More Egocentric? Age Differences in the Impact of Retrospective Self-Experience on Interpersonal Emotion Intensity Judgment"

_behavsci, 2024, doi:10.3390/bs14040299_

Round 1

Reviewer 1 Report

Comments and Suggestions for Authors

This study can be evaluated as a challenging study that examines the interpersonal emotion judgment from the perspective of anchoring with retrospective self-experience as the reference point. On the other hand, there are many variables being measured, and it is considered that the explanation and analysis of the process by which these variables are related to the interpersonal emotion judgment are not sufficient.

Intorodction

1. The concept of this study is complex as it measures many variables including retrospective self-experience (autobiographical memory), anchoring, processing speed, inhibition, and working memory. You should explain how these are related to the information processing for emotion judgments of others with using figure so that the reader can correctly understand.

2. The following hypotheses (Line 94 - 99) may not be appropriate. The positivity effect would be observed in the recall of autobiographical memories for older adults. Therefore, they are expected to recall more positive experiences, resulting in stronger anchoring (more egocentric judgment) in positive emotion judgment of others. Young adults have negativity bias, and thus can be predicted to show a tendency toward egocentric judgment in negative emotions.

In addition, you should make hypotheses of the age difference based on previous research of the anchoring effect and aging.

“Moreover, previous research has indicated that older adults display a preference for positive information rather than negative information in attention and memory (Mather & Carstensen, 2005). Additionally, older adults are more prone to underperforming than younger adults when judging negative emotions (Ngo & Issacowitz, 2015). Therefore, the present study speculates that, compared to younger adults, older adults exhibit more egocentric tendencies in the judgment of negative emotions than positive emotions.”

Results

3.  In the manipulation check (Line 264), a two-way ANOVA with age group and anchor group for the emotional intensity of the retrospective self-experience should be conducted to distinguish the effects of age difference on each of the self-emotional intensity of the retrospective self-experience and anchoring. 

4. Since the relation between variables is important information to better understand the results, please provide tables on correlations for each age group. You may add this information to the supplemental file.

Discussion

5. Please check the following statement as it may be incorrect.

Specifically, the results for anger exhibited the greatest age difference, with older adults outperforming younger adults in anchoring effects.(Line 374-375)

6. Line 357-359: “This finding suggested that older adults' judgments of others' emotions were disproportionately impacted by their retrospective self-experience, even when they were asked to provide objective estimates.”

Why do the results of this study suggest that the judgment of older adults is "disproportionately impacted"? What this study may have revealed is that when judging the emotion of others, older adults may be more likely than younger adults to use their own experience as a reference point for their judgments. Additionally, the younger adults were shown to be more egocentric in their judgments of anger. It is reasonable to assume that age differences in judging emotion of others may differ depending on the emotional valence.

Reviewer 2 Report

Comments and Suggestions for Authors

1.The existing body of research has consistently highlighted the challenge faced by healthy elderly individuals in recognizing emotions such as anger and sadness. This difficulty is posited to contribute to an overall decline in emotion recognition, irrespective of factors such as perceptual ability, processing speed, fluid IQ, and basic facial expression recognition. Interestingly, there appears to be a contradiction between the findings of this study and those of related research concerning the relationship between interpersonal emotion recognition and cognitive ability. A comprehensive and rational explanation is sought to reconcile these seemingly conflicting claims.

Sullivan, S., & Ruffman, T. (2004). Emotion recognition deficits in the elderly. International Journal of Neuroscience, 114(3), 403-432.

2. Echoing the assertions of the present study, previous research has indicated a correlation between lower susceptibility to regret and better overall health in the elderly population. However, an intriguing nuance emerges when examining the response of healthy older adults when confronted with outcomes that deviate from reality; in such cases, they tend to interpret these instances as counterfactual events, leading to reduced responsiveness to counterfactual outcomes. It is crucial for the authors to expound on the influence of their hypothetical facts on the experimental results, especially considering that the experiments in this study are designed around testing hypotheses based on hypothetical facts.

Brassen, S., Gamer, M., Peters, J., Gluth, S., & Büchel, C. (2012). Don't look back in anger! Responsiveness to missed chances in successful and nonsuccessful aging. Science, 336(6081), 612-614.

Strong Points

Comparing differences in anchoring tasks between older adults.

Cons

There are contradictions in the conclusions drawn from the test results.

The premise is that the test determines whether the difference being compared is due to chance or not, and does not represent the strength. Furthermore, the failure to find a significant difference does not indicate that there is no difference.

With that in mind, this study concludes that older adults are more susceptible to the effects of anchoring, but we do not believe there is a significant difference between older and younger adults in high anchoring. The same is true between older and younger adults at low anchoring.

Additionally, a related study [1] has reported that healthy elderly people have difficulty recognizing anger and sadness. This is argued to result in a decline in emotion recognition, independent of perceptual ability, processing speed, fluid IQ, and basic facial expression recognition. Regarding the relationship between interpersonal emotion recognition and cognitive ability, the claims of this study and those of related research appear to be contradictory. I'm looking for a rational answer. When healthy older adults are faced with an outcome that is different from reality, they can view it as a counterfactual event [2]. This counterfactual event refers to the hypothetical scenario we conducted this time. Therefore, healthy older adults are less responsive to counterfactual outcomes. The experiments in this study test hypotheses based on hypothetical facts. You should mention the effect that your hypothetical facts have on the experimental results.

[1] Sullivan, S., & Ruffman, T. (2004). Emotion recognition deficits in the elderly. International Journal of Neuroscience, 114(3), 403-432.

[2] Brassen, S., Gamer, M., Peters, J., Gluth, S., & Büchel, C. (2012). Don't look back in anger! Responsiveness to missed chances in successful and nonsuccessful aging. Science, 336(6081), 612-614.Brassen, S., Gamer, M., Peters, J., Gluth, S., & Büchel, C. (2012). Don’t look back in anger! Responsiveness to missed chances in successful and nonsuccessful aging. Science, 336(6081), 612-614.

Reviewer 3 Report

Comments and Suggestions for Authors

The authors had written a good article. Here are some tips to improve the article.

The introduction needs better organization. To improve the logical flow and readability, related studies and findings should be grouped.  

In the method section, information should be provided on how to achieve emotion categories in the productive anchor and balanced goal scenarios. For instance, the criteria used to select scenarios that represent each category of emotion equally or the process of ensuring an equal distribution of positive and negative emotions can be mentioned. Additionally, a clearer timeline of study steps is needed, specifying the duration of each task, the order in which participants completed them, and any breaks or intervals between tasks. 

The discussion section requires some elaboration. Firstly, the results that are consistent with previous studies on perspective-taking and emotional judgment should be explained in more detail. Secondly, regarding the age-related tendency observed in the anchoring effects, further clarification is needed on the potential implications of these findings. The practical implications or potential applications of these findings in real-world settings should be explained. Lastly, in the discussion of age differences in the effect of anchoring anger, the possible reasons why older adults perform better than young adults in the effects of anger inhibition should be discussed, such as decreasing the daily experience of anger and changing sensitivity to anger with increasing age. In the end, the references are also old, please use updated references.

Round 2

Reviewer 1 Report

Comments and Suggestions for Authors

I reviewed a previous version of this manuscript and believe that the authors have done a good job to address my points. However, please add information on the following point:

1. Please provide f, p, and effect size for the non-significant main effect and interaction in the Manipulation check (Line 319).

Reviewer 2 Report

Comments and Suggestions for Authors

I have verified that the authors have followed the comments and made the appropriate corrections.

Author Response

For research article:

The manuscript titled “Who Will Be More Egocentric? Age Differences in the Impact of Retrospective Self-Experience on Interpersonal Emotion Intensity Judgment”. The manuscript ID is behavsci-2901853

Comment 1: I have verified that the authors have followed the comments and made the appropriate corrections.

Reply: Thank you for confirming that we have addressed your comments and implemented the necessary revisions.